# The Fate of Lipid-Coated and Uncoated Fluorescent Nanodiamonds during Cell Division in Yeast

**DOI:** 10.3390/nano10030516

**Published:** 2020-03-12

**Authors:** Aryan Morita, Thamir Hamoh, Felipe P. Perona Martinez, Mayeul Chipaux, Alina Sigaeva, Charles Mignon, Kiran J. van der Laan, Axel Hochstetter, Romana Schirhagl

**Affiliations:** 1Department of Biomedical Engineering, University Medical Center Groningen, University of Groningen, Antonius Deusinglaan 1, 9713 AV Groningen, The Netherlands; drg.armorita@gmail.com (A.M.); thamirhamoh@gmail.com (T.H.); felipeperona@gmail.com (F.P.P.M.); mchipaux@gmail.com (M.C.); aosigaeva@gmail.com (A.S.); charles.antoine.mignon@gmail.com (C.M.); kiranvanderlaan@gmail.com (K.J.v.d.L.); 2Department of Dental Biomedical Sciences, Faculty of Dentistry, Universitas Gadjah Mada, Yogyakarta 55281, Indonesia; 3Division of Biomedical Engineering, School of Engineering, University of Glasgow, Glasgow G12 8LT, UK; axel_hochstetter@web.de

**Keywords:** fluorescent nanodiamonds, cell division, yeast

## Abstract

Fluorescent nanodiamonds are frequently used as biolabels. They have also recently been established for magnetic resonance and temperature sensing at the nanoscale level. To properly use them in cell biology, we first have to understand their intracellular fate. Here, we investigated, for the first time, what happens to diamond particles during and after cell division in yeast (*Saccharomyces cerevisiae*) cells. More concretely, our goal was to answer the question of whether nanodiamonds remain in the mother cells or end up in the daughter cells. Yeast cells are widely used as a model organism in aging and biotechnology research, and they are particularly interesting because their asymmetric cell division leads to morphologically different mother and daughter cells. Although yeast cells have a mechanism to prevent potentially harmful substances from entering the daughter cells, we found an increased number of diamond particles in daughter cells. Additionally, we found substantial excretion of particles, which has not been reported for mammalian cells. We also investigated what types of movement diamond particles undergo in the cells. Finally, we also compared bare nanodiamonds with lipid-coated diamonds, and there were no significant differences in respect to either movement or intracellular fate.

## 1. Introduction

The fluorescent nanodiamonds (FNDs) are promising long-term biolabels due to their unprecedented photostability [1,2,3]. They can host fluorescent defects such as the nitrogen vacancy (NV) center. These centers can be excited with a green laser (532 nm) and emit red fluorescence (a broad peak above 600 nm). NV centers occur naturally in nanodiamonds from high-pressure high-temperature (HPHT) synthesis, but their numbers can be increased by irradiation in several different ways. These increase the number of color centers and thus their fluorescence intensity [4]. Possibilities include irradiation with silicon ions [5], helium ions [6], or electrons [7,8]. For biological applications, the excellent biocompatibility of fluorescent nanodiamonds is also crucial [9]. In several previous studies, FNDs are introduced in mammalian cells and have shown no negative effects [10,11,12]. From these studies, it is known that mammalian cells passively take up FNDs in different ways depending on the cell type and exact conditions. The most reported uptake path is endocytosis. When nanodiamonds are endocytosed they are engulfed in an endosome and eventually escape from it [13]. Besides, little is known so far about the behavior of these FNDs after uptake or about what happens to them during cell division. Although most studies are limited to short times where division does not occur, there is a small number of articles on the behavior of FNDs during cell division in mammalian cells [14,15].

In this study, yeast cells were used as model organism. We have shown before that FNDs can be brought inside these cells [16]. For yeast cells, the uptake mechanisms of nanoparticles are unknown. As they are covered with a thick cell wall, the uptake is also a lot more artificial than in mammalian cells and there is probably no natural mechanism that the uptake is comparable to. To achieve uptake, two protocols have been established: One option is to permeabilize the cell wall, which allows the diamond particles to enter. It has also been shown in a previous work that the cells could proliferate after being treated with FNDs [16,17]. Another option, which we used here, is to remove the cell wall entirely. This method allows the diamonds to enter and regrow the cell wall [18]. Here, we investigate for the first time what happens during and after cell division in yeast. This is especially interesting for yeast cells, because the division is asymmetric. Asymmetric division can manifest itself in different ways, for instance, different cell content. In yeast cells, division results in differently sized mother and daughter cells. Before cell division, a diffusion barrier keeps molecules (in this case, FNDs) in the membrane of the mother and prevents them from entering the membrane of daughter cell. In the FNDs, other particles or molecules only leak into the daughter cells if they detach from the membrane in the mother cells or if the diffusion barrier becomes permeable [19]. This mechanism is in place to protect daughter cells from harmful substances like aging factors [20].

Compared to other organisms, yeast cells have several advantages for this kind of research. They are a relevant model to study the aging process, and they are widely used in biosynthesis and in food industry [21,22,23]. They are undemanding in cultivation and allow for easy genetic and molecular modifications [21].

Cell division is a very important step in the aging process of yeast cells. When investigating the transfer of FNDs during cell division, in principle, there are four possible outcomes after cell division:

FNDs could remain (preferentially) with the mother, for example, because they are regarded as harmful by the cell (see Figure 1a).FNDs might (preferentially) move into the daughter cells (see Figure 1b).FNDs might be excreted (see Figure 1c).FNDs might end up randomly distributed between both mother and daughter cells (see Figure 1d).

The goal of this article is to determine which one of these possibilities is the case for nanodiamonds. To answer this question, we used FNDs in yeast cells using the spheroplasting process [18,24] (i.e., removing the cell wall) and we followed them during cell division. We investigated here which of the four possible outcomes occur and in what frequency.

## 2. Materials and Methods

### 2.1. Diamond Starting Material

#### 2.1.1. Bare Particles

Throughout this article we used fluorescent diamonds with a hydrodynamic diameter of 70 nm (FND_70_) from Adamas Nanotechnology (Raleigh, NC, USA). They have a relatively broad size distribution and irregular shape [25]. According to the vendor, these particles are irradiated with an electron beam at 3 MeV to 5 × 10^19^ e/cm^2^ fluence followed by high temperature annealing above 600 °C under vacuum for 2 h [26]. The NV content was measured by the manufacturer by electron paramagnetic resonance to be approximately 2–2.5 ppm. This means each particle hosts approximately 300 nitrogen vacancy centers. We measured their fluorescence spectrum (see Appendix A) on a Thermo Fisher Varioskan microplate reader, with excitation wavelength at 532 nm, and analyzed it as shown by Fu et al. [27]. We found that the particles contain almost exclusively NV^−^ centers, which is also in line with what others found for similar particles [28]. With our homebuilt confocal microscope we can detect ~1,000,000 counts per second for a single particle. This was determined in previous works where we spread particles evenly on a surface (confirmed by SEM) and measured the counts [25]. As they undergo a cleaning process in oxidizing acid, their surface is oxygen terminated and electronegative with zeta potential of −16 ± 1 mV.

#### 2.1.2. Coated Particles

For facilitating FNDs uptake in yeast cells, a liposome kit (Sigma, Zwijndrecht, The Netherlands) has been used as coating material. This kit contains 63 µmol L-α-phosphatidylcholine and 18 µmol stearylamine. After the coating process, the zeta potential value of FNDs becomes electropositive (36 ± 3 mV) [18]. To prepare FNDs coated with lipids (FND-lip), 2 µg mL^−1^ of FND solution was added into liposomes and was mixed by vortexing for 30 s.

#### 2.1.3. Particle Characterization

Characterization of diamond and lipid-coated diamond particles (FND-lip) has been performed in a previous study [18]. There we characterized several properties for these particles. The findings are summarized here shortly. No significant differences in size between FND-lip and FNDs were observed. There we found that both FND-lip and FNDs are colloidally stable in water (PdI < 1). To further characterize the particles, we performed an analysis of the zeta potential. To this end, 4 µg/mL liposome-coated FNDs were diluted in sterile deionized water and 1 mL of the solution was injected into the cuvette and 4 µg/mL FND70 was used as control. The measurements were performed with a Malvern Zetasizer Nanosystem (Malvern, Cambridge, UK). All the measurements were set in 25 °C. Each measurement takes ~2 min. The zeta potential measurement showed that the 70 nm FNDs were electronegative (−15.73 ± 0.89 mV). After adding liposome, the particles became electropositive (35.67 ± 2.64 mV) [18]. Cryo TEM (recorded with Tecnai, Oregon, USA) revealed that the thickness of the lipid layer on diamond particles was 4.8 ± 1.2 nm [18]. It is also apparent from this previous study that the FNDs are actually coated by lipids. Performing optically detected magnetic resonance measurements (as routinely used in the field) on FNDs and FND-lip did not reveal any significant differences [18]. Although FNDs generally are known to have excellent biocompatibility [9,29], a very small decrease in metabolic activity has been reported for yeast cells and FND-lip [18].

### 2.2. Fluorescence Nanodiamond Particles Uptake

*Saccharomyces cerevisiae* BY4741 and HXT6-GFP strains were used as model organisms. According to the Saccharomyces genome database, the wild type strain BY4741 was used as a parent strain for an international systematic *S. cerevisiae* gene disruption project. Thus, it was chosen here for its broad use. These wild type cells were used for tracking intracellular movement of FNDs. The HXT6-GFP strain was used for quantifying FNDs. This modified strain expresses Hexose transporter 6 (glucose transporter) with green fluorescent protein (GFP) in the cell membrane, thus allowing imaging of the cell boundaries. Both cells were grown in synthetic dextrose (SD, Formedium, Norfolk, UK) complete medium until midlog phase (OD_600_ = 1.05). The spheroplasting protocol was modified from Karas et al. [24] and was performed to get the FNDs inside cells. The adaptation from the original protocol was that after spheroplasting they put the spheroplast on specific medium and we did not do that. In the spheroplast protocol, the cell wall is removed entirely from the yeast cells to create spheroplasts. To obtain these spheroplasts, the cells were washed with sterile demineralized water and centrifuged for 5 min at 2500× *g* at 10 °C. The supernatant was discarded, and 20 mL of 1 M D-sorbitol was added to the cells. The cells were again centrifuged for 5 min at 2500× *g* at 10 °C. After discarding the supernatant, 20 mL of SPEM (consisting of 1 M D-sorbitol, 10 mM EDTA, and 10 mM sodium phosphate) buffer was added followed by 40 µL zymolyase 20 T (Amsbio, UK) and 30 µL *β*-mercaptoethanol (Sigma, Zwijndrecht, The Netherlands). Cells were incubated at 30 °C while shaking at 75 rpm for 30 min. Twenty milliliters of 1 M D-sorbitol was added to stop the spheroplasting process, and the cells were centrifuged for 5 min at 1000× *g* at 10 °C. After the supernatant was discarded, 2 mL of STC (1 M sorbitol, 10 mM TrisHCl, and 10 mM CaCl_2_ and 2.5mM MgCl_2_) buffer was added and the mixture was incubated for 20 min at room temperature. In the end, 50 µL of 2 µg/mL FNDs at a size of 70 nm were added to the 200 µL yeast spheroplast suspension, followed by 5 min incubation at room temperature. Finally, the treated yeast cells were put in SD complete medium supplemented with 1 M D-sorbitol for 1 h at 30 °C to regrow their cell wall.

### 2.3. Immobilizing Yeast Cells

To monitor single cells during and after cell division they were immobilized using the following protocol; glass-bottom dishes with 4 compartments were coated with 0.1 mg/mL concanavalin A (Sigma, Zwijndrecht, The Netherlands). The coating process was followed by a washing step with sterilized demineralized water and a drying step in a 37 °C incubator. After the coated dish dried, 300 µL SD medium and 4 µL of cell suspension (strain BY4741, approximately 2.4 × 10^7^ cells/mL) with internalized FNDs from the previous step were added in each compartment and the dish was sealed by parafilm to avoid evaporation of the medium.

### 2.4. Equipment

Imaging was performed on a home-built confocal microscope operating with a 532 nm excitation laser. The confocal microscope is similar to what is typically used in the diamond magnetometry community [30,31]. Below we shortly describe the most important specifications. A detailed description including a drawing (Appendix A) and a list with all the parts of our equipment can be found in the Appendix A. We have a homebuilt system because it allows for flexibility to perform diamond magnetometry. However, this functionality was not used in this article, and the measurements could have also been performed on a commercial system with similar capabilities. For detection, our instrument has an avalanche photodiode implemented for detection, which is capable of single photon counting. The fluorescent counts we receive for 70 nm diamond particles are typically ~1,000,000 per second for a single particle. These values are close to what we expect for this number of NV centers per particle. The instrument has built-in microwaves (which we do not use in this article) and uses sensitive detection with avalanche photodiodes. The set-up is equipped with a green laser at 532 nm, and we have the ability to track particles in 3D. The sample stage is designed in a way that allows for standard glass-bottom petri dishes to be measured. For the measurement, the sample suspension was dropped onto a microscope cover slide and evaporated at room temperature. The instrument was set to −12 dBm of microwave power and 1 mW of laser power. One-hundred repetitions were performed to obtain a sufficient signal-to-noise ratio [18]. To better identify the cells, the confocal microscope is equipped with a bright-field microscope, which is used to collect images simultaneously. Bright-field illumination is achieved with a 470 nm Fiber coupled LED supplied with T-Cube LED Driver. The images are collected using a Compact USB 2.0 CMOS Camera from Thorlabs, and an Olympus PLN 4x objective to focus the blue light with NA 0.1.

### 2.5. Tracking FND Movement during Cell Division

To separate the FND signal from other fluorescence, a 550 nm long-pass filter was used. A signal above 550 nm was attributed to the FNDs. It is also possible to use a filter above 600 nm (or higher), but there is a trade-off. If one uses a higher filter, the technique is more specific for ND. However, one also loses part of the signal and thus sensitivity. Therefore, if the background is comparably low, it is possible to choose a lower wavelength filter to gain sensitivity. We detect on average 90,000 ± 10,000 counts per second for the background of the cell, whereas the FNDs are 1,000,000 ± 500,000 counts per second (for control images without particles see Appendix A). A laser power of 60 μW at the laser power output was chosen to limit potential damage to the cells from high laser power. First, we scanned an area (50 × 50 µm field of view) with cells. Then, we identified diamonds by observing stability of their fluorescence intensities, as diamonds are not bleaching. Usually, we observe a particle for ~10 min. If the fluorescence does not drop, it is most likely a diamond. Images were acquired every hour. Light intensity was measured using an Olympus UPLanSApo40x NA = 0.95 air objective and an Avalanche photodiode (SPCM-AQRF-15-FC) in single photon counting mode. Simultaneously, bright-field time series images were recorded continuously to give a better view of cell division. Confocal images were processed in FiJi software using specific plugins [31]. Deconvolution was performed to get clearer particle locations and lower background using Diffraction point spread function (PSF) 3D and Iterative deconvolve 3D plugins.

### 2.6. FND Quantification during Cell Division

While following the FNDs during cell division (using the above mentioned confocal microscope in Appendix A), FND quantification was performed after re-growing the cell walls. Four microliters of yeast spheroplast suspension contained approximately 9.6 × 10^4^ cells (strain HXT6-GFP) that were fixed using 1% paraformaldehyde in PBS buffer of pH 7.4. The cell suspension was put between a glass slide and the cover glass and was imaged with Zeiss LSM 780 confocal laser scanning microscope (Zeiss, Oberkochen, Germany). FNDs were imaged at excitation/emission wavelength 561/650 nm and GFP was imaged at 488/525 nm. A homemade FiJi program was used for determine the number of particles that have been ingested by the cells [16] before and after cell division. To this end, a specific, custom-made FND quantification plugin was used to approximate the amount of internalized FNDs. The analysis was divided into three phases: Cell Selection, Masking, and Particle Analysis. During the first phase, images are visually inspected and random cells are selected. The images were composed of several slices (Z-stacks), and the cellular region was defined in all the three dimensions. In the horizontal plane, the selection considered an area containing only the cell of interest. In the height, the first and last slices containing the cell were identified. As a result, the first phase defines a volume that holds only the cell of interest. In the masking phase, that volume is molded to resemble the shape of the cell. The GFP signal (staining the cell membrane) is converted to binary using the Isodata algorithm to calculate the threshold [32], and the cell’s perimeter is detected in every slice. To avoid counting particles on the surface, the program excludes the outer micrometer of the volume. (As a result, the program slightly underestimates the real number of particles.) In the third step, a special function of Fiji analyses the particles, which are found in a selected region. Applying this function to the masked image, it is possible to directly obtain the number of objects (connected positive pixels) in the specified region. A threshold is used to separate the background light from the signal emitted by the FNDs. Every pixel with intensity less than the threshold is assumed as background and set as black, whereas every pixel with an intensity greater than or equal to the threshold is assumed as part of a particle. To find an adequate value for this parameter, the image was visually inspected and different values were probed. Finally, we chose the lowest possible value, which gives zero for a negative control image. In the end, this method gives two important values: the number of objects and the number of particles. An object here is defined as the amount of adjacent FND positive pixels. This means an object can be composed of a single diamond or can be an aggregate of multiple diamond particles. The difference between the number of objects and the number of particles reveals the aggregation status of the sample in the intracellular environment (rather than in a test tube as in DLS for instance).

### 2.7. Single Particle Tracking

To better understand the intracellular processes a nanodiamond can be subjected to, we performed single particle tracking. This was done for both uncoated (bare FNDs) and lipid-coated nanodiamonds (FND-lip) to study the impact of enclosing nanodiamonds in liposome vesicles inside the cell. As a liposome is larger (the thickness of liposome on FND surface was determined by TEM to be 4.8 ± 1.2 nm [18]) than an uncoated nanodiamond, we would expect it to move slower, according to Fick’s laws. If the particle is, on the other hand, actively moved to a specific compartment, a directed motion can be observed. Using a home-built confocal microscope (for details see Appendix A), the sample was excited with a 532 nm laser. Matching the confocal image with a bright-field image was done to confirm if the diamond is inside the cell. After finding a particle, it was tracked for ~45 min. The custom-made software recorded 2 × 2 µm images of the particles throughout the tracks to make sure we are following the same particles. By applying a Gaussian fitting of the intensity profiles, the software calculates the position of the particle center. Cells were put in the coated glass-bottom dishes and, to keep the cells alive, we performed the experiments only for 45 min. This time was chosen conservatively after several preliminary experiments were cells died after longer exposure times. Temperature in the samples was ~25 degrees.

Recording images of a particle took 1.8 s per image. In total, we performed 1500 repetitions. To interpret these trajectories, mean square displacement (MSD) curves [33] were used. This value provides a measure on how far a particle travels within a certain time period (τ). These trajectories were divided into subtracks using a rolling window (200 repetitions) to gain more information about each segment of the trajectory.

The MSD curves were calculated from the coordinates of the recorded trajectories. The typical logarithmic scaling of a graphical representation of the MSD versus the time interval allows differentiating between three typical cases: purely random motion (diffusion), ballistic motion overlaying random diffusion, and confined diffusion. For all three cases we have simulated trajectories and calculated the corresponding MSD curves (see Figure 2), using Python.

For the simulated trajectories, a random generator drew 2000 displacements with varying step length between −1 and 1. To better simulate the actual movement of the microscopic particles, we assumed that we do not record every single displacement but only every third, by cumulating three displacements into one. This also ensures that the distribution of step sizes and displacements follow a normal distribution. For the ballistic motion, one additional drawing with a directed motion (only positive values for x and y) was added to the three random displacements. A confined displacement trajectory was generated like the random motion with an additional step: whenever the trajectory of the particle left a predefined area (here: 30 by 30 microns), the particle was reflected into the area—according to a particle hitting the wall of its confinement. However, this means that the particle becomes faster when it is reflected from the border, which can be seen in the slightly higher slope of the left side of the graph for confined motion in Figure 2 (green). The full Python script used to generate the trajectories and to calculate the mean squared displacements can be found in the Appendix A.

Simple diffusion is shown in red—the particle performs a random walk, slowly moving away from the original position, and can theoretically explore every point of the volume. Similar behavior can be observed for a particle floating in suspension. The green trajectory illustrates confined diffusion—the particle is walking randomly within a certain limited volume (e.g., a vesicle, a cytoskeletal “cage”, or even an entire cell) and cannot leave it, no matter how much time has passed. The blue trajectory shows the case of simple diffusion combined with directional motion. The particle has a preferred direction, in which it moves with a certain speed (e.g., being transported along the cytoskeleton by molecular motors). At the same time, the diffusional component results in deviations from the shortest straight line between the initial and the final positions.

At high τ, less data points contribute to the calculation, and thus the MSD calculation has a larger statistical uncertainty. To compensate for this, we used only the first 75% of the resulting curves for fitting.

### 2.8. Statistical Analysis

All statistical analyses were performed with GraphPad prism with 95% confidence interval. One-way analysis of variance (ANOVA) was performed to analyze significance between particle uptake experiments. Correlation assessment was performed for comparing cell size and number of particles, and the Student *t*-test was used to analyze single particle tracking parameters inside cells.

## 3. Results

### 3.1. FNDs Movement during Cell Division

The particles inside cells have been followed between 0 and 6 h after the particle uptake process. These times were used because according to Broach et al. [34], yeast cells proliferate for about 6 h in fermentable carbon sources (in this experiment, SD medium supplemented with 2% of glucose). Hu et al. used the same time span [35]. The microscopic images resulting from this experiment are shown in Figure 3. Observing the movement of FNDs in living cells during cell division, we found three different cases of what the cell does with the particles. To display the different cases that we differentiate, we show three cells that behave differently under the same conditions. There are several possible explanations for the biological variety leading to these different behaviors. The most important difference is probably that the cells are in different stages of their cell cycle when the experiment is started. Additionally, there are genetic differences as well as differences in the metabolic state of cells.

### 3.2. FND Quantification after Cell Division

As mentioned earlier, there exists biological variation between cells. Therefore, we measured 100 cells per condition to get a more complete picture of what happens and to obtain statistics. To analyze the distribution of FNDs over mother and daughter cells, a particle counting protocol [16] was performed before and after cell division (see Figure 4).

Comparing Figure 4a,b, we see that the overall number of particles/objects per cell is smaller after division than before. Thus, we surmise that for both particle types a large proportion of diamond particles is excreted from the cells. Excretion in yeast has been observed for diamond [17] as well as for other materials. The process has been described to happen when the cells are not in a balanced condition. This can be, for example, when the cells are transferred to a medium lacking a required compound. As a consequence, they are producing unusually large amounts of amino acids [36]. We do not find a significant difference in the number of objects per cell and number of particles per cell between FND-lip and bare FNDs. This is the case both before cell division and after cell division (*p* value < 0.05). Correlation analysis was performed for both groups, and the results show no correlation between cell size and number of particles in FND-lip (R^2^ = 0.0007) or FND (R^2^ = 0.0684). This finding suggests that particles move randomly during cell division.

Based on Jorgensen et al. (2002) [37], the volume of mother cells in yeast is approximately 420–820 nm^3^. We categorized the cells based on the volume. Cells with a volume greater than 420 nm^3^ were assigned to the mother group, whereas cells with smaller size are counted in the daughter group. Figure 5 shows the distribution of particles in mother and daughter during the measurement. We divided groups based on cell volume and the number of particles the cells contain.

### 3.3. Single Particle Tracking

In this study, we wanted to observe if the presence of liposomes has an effect on the particle behavior inside living cells. A control experiment with FNDs has been performed as a comparison with FND-lip. We compared the tracked pattern of particle movement in 3-dimensional trajectories and mean square displacement for both FNDs (Figure 6) and FND-lip (Figure 7). Due to the small size of yeast and the limited optical resolution, it is tricky to conclude whether a particle is inside or outside of a cell. However, we also observed the movement of particles in the medium alone and in glycerol. The medium alone is much less viscous than the cellular content, and the movement of the particles is too rapid to be tracked with our set-up. FNDs moving freely in glycerol explore substantially larger volumes than the particles tracked in the experiments with yeast cells (Appendix A). We observe the displacements of 10^2^–10^3^ μm^2^ and largely unconstrained (simple) diffusion, with the values of α being 0.58 on average and not dropping below 0.32 (Appendix A). Thus, the confined movement we observe in this set of experiments is much more consistent with a movement inside cells. For a comparison with the simulated data see Appendix A. The 3D trajectories suggest that liposome coating has only little effect on the movement of the particles. Both coated and uncoated particles were in a confined diffusion type of movement. Comparison of MSD curves showed that the displacement area was not statistically different between the particles with and without lipid coating.

A statistical analysis has been done to compare the displacement between bare and lipid-coated diamonds. It indicates that there is no significant difference between the two groups with *p*-value 0.1131 (Figure 8a). Alpha indicates the how freely a particle moves. The higher the value of alpha, the larger is the volume available to the particle. The results showed a significant difference in alpha (*p* value 0.007) between the two groups (Figure 8b). The presence of liposomes surrounding FNDs facilitates aggregate formation and thus results in change in the volume and limitation of the movement. This difference is reflected in a lower alpha value in FNDs-lip (Figure 8b). The diffusion coefficient Figure 8c shows no significant difference (*p* value 0.972) between uncoated and FND-lip.

## 4. Discussion and Conclusions

Here we provide the first analysis of the fate of FNDs during and after yeast cell division. We found that there are multiple routes that the particles can take and in total we observed three different fates of particles during cell division. For bare FNDs, 14.9% of particles stayed in the mother cell, compared to 21.6% for the FND-lip (Figure 3a). Twenty-one percent of the bare FNDs and 28.4% for the FND-lip ended up in the daughter cells (Figure 3b). Finally, 98.35% of particles for FNDs and 98.38% of the particles for FND-lip were most likely excreted from the cell (Figure 3c).

Surprisingly, the distribution of particles between mother and daughter cells (result in Figure 5) shows that more daughter cells in both groups (both bare FND and FND-lip) contain more particles than mother cells (Figure 5). There is a large body of literature available where nanoparticles interact with yeast. However, these articles are almost exclusively concerned with the toxic effect that nanoparticles could have on yeast. The distribution between mother and daughter cells is so far investigated for heat-induced protein aggregates (80 nm) [38]. The size of FNDs used for this study (70 nm) is comparable to these particles. However, in the case of FNDs, we did not observe more particles being retained in the mother cells, which was typical for protein aggregates in dividing yeast cells. This discrepancy can be explained in different ways.

A first explanation is related to the size of the particles. For heat-induced protein aggregates, passive aggregate formation, fusion, growth, and exclusion from the bud was observed [38]. As the protein aggregates grow, they can reach sizes of 600 nm [39]. We do not see this for FNDs. In this case, one might assume that nonaggregated FND particles are substantially smaller, thus it is easier for them to cross the bud neck and end up in the daughter cell.

On the other hand, there are a number of models that suggest the impact of certain active control mechanisms on the particle distribution in the dividing cells. The protein aggregates are thought to be retained in mother cells due to their tethering to actin cables, retrogradely moving to the mother cell [40], attachment to the nucleus and the vacuole [41], or to the surface of endoplasmic reticulum and anchored mitochondria [42]. The first model suggests active transport of the aggregates out of the bud, whereas the second and third models explain the asymmetric distribution of the aggregates by general decrease in their mobility. If any of these models are true, it might mean that FNDs simply lack certain features of protein aggregates and are thus “invisible” for those mechanisms of active control. Molecules that are adsorbed on the diamond surface could also influence how they interact with the cells. It has been shown for other nanoparticles that the protein corona can dictate the interaction with cells. Drug-loaded nanodiamonds, on the other hand, behave more like bare diamond particles in mammalian cells than drug molecules alone. The fact that FND and FND-lip behave similarly, despite the different surface chemistry, indicates that here the behavior might also be dictated by the diamond particle. It can be useful that both mother cells and daughter cells contain diamonds. This way it is possible to observe and track both. Additionally, FNDs might also be considered “safe” for the cells and are thus not actively hindered from entering daughter cells.

The spheroplasting process might also affect the fate of FNDs by giving a stress condition to the cells after being treated by zymolyase affect cell wall integrity [43], which is used for intracellular transport [44]. In this case, changing of cell wall components (especially actin) due to spheroplasting process will affect particle movement during cell division. Both yeast and spheroplasts have a cell membrane; in spheroplasts, only the cell wall is removed. When FND-lip (which have a positive zeta potential) are taken up by the cells, they bind to negatively charged components of the plasma membrane. Another study also showed that positively charged particles can be ingested by the cells better than negatively charged particles [45].

Finally, we also observed and quantified movement of particles during cell division. We mainly find confined diffusion for both coated and uncoated particles without showing significant differences between displacement and diffusion coefficient. On the other hand, we have noticed a significant difference in alpha. This is due to the fact that coated particles have the tendency to aggregate. Thus, this movement is more confined.

The results from this study might in the future be useful for labeling yeasts with diamonds or magnetometry in yeast cells. Additionally, this work might be interesting to compare with how other nanoparticles behave during cell division.

## Figures and Tables

**Figure 1 nanomaterials-10-00516-f001:**
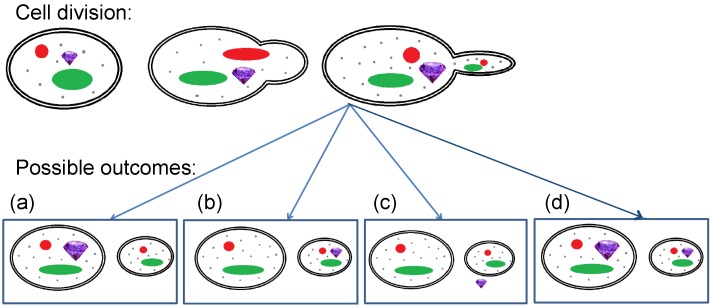
In asymmetric cell division, a yeast cell produces a smaller daughter cell. Except for the size, the daughter cells are similar to the mother cells. Both of them have nuclei (red), a vacuole (green), and other organelles (gray). When fluorescent nanodiamonds (FNDs) (purple) are inside the cells during division, they can (**a**) stay in the mother cell, (**b**) move to the daughter cell preferentially, (**c**) being excreted by the cells, or (**d**) being equally distributed.

**Figure 2 nanomaterials-10-00516-f002:**
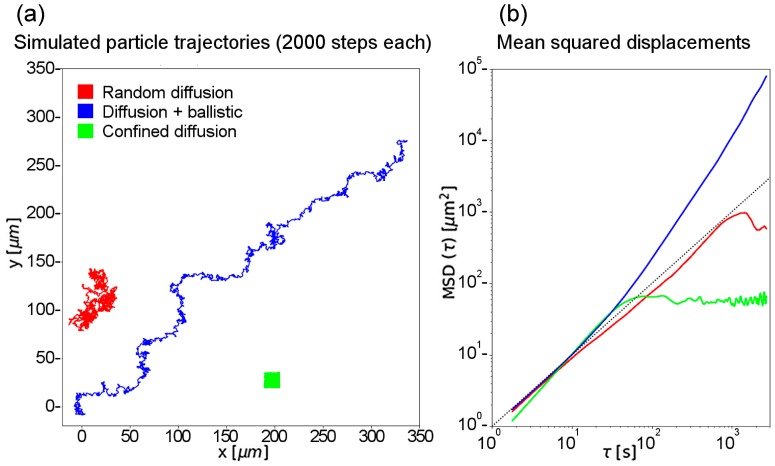
Three possible types of the particle movement generate three distinct trajectory types: random (red), ballistic (blue), and confined (green). The trajectories (**a**) represent the movements a particle may execute over 2000 steps: random (red), random diffusion with overlying ballistic motion (blue), and confined (green), wherein the trajectory of the particle is limited by a predefined border (here: 30 by 30 microns) that reflects the particle. (**b**) The mean squared displacements of the trajectories.

**Figure 3 nanomaterials-10-00516-f003:**
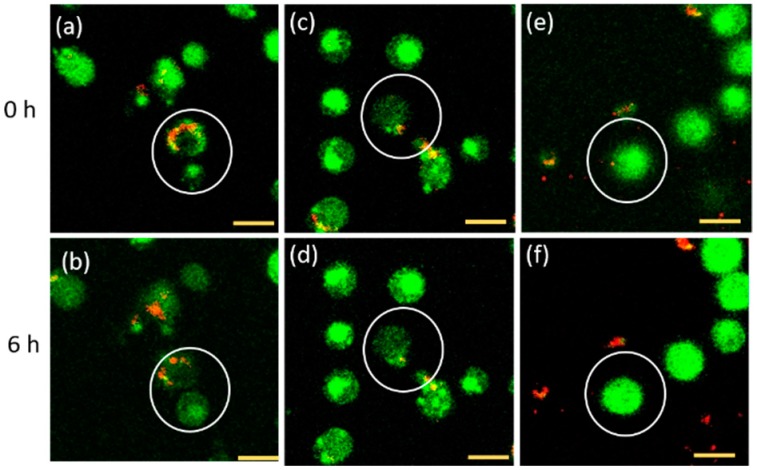
Different cases for particle fate during cell division. The red particles show the position of the fluorescent nanodiamonds, whereas green color indicates the cell stained with GFP. The first row shows images at 0 h while the second row shows the same cells at 6 h. In panels (**a**,**b**), the particles have remained in the mother cells over time. In panels (**c**,**d**), the particle has moved to the daughter cell. In panels (**e**,**f**), the particle that was in the cell at time zero has completely moved out of the cell after 6 h. The scale bars are 2 micrometers long.

**Figure 4 nanomaterials-10-00516-f004:**
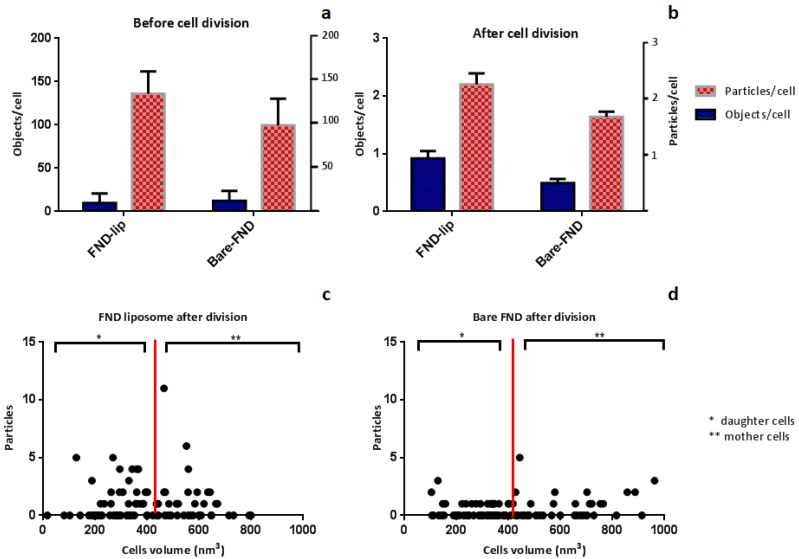
Comparison of number of objects and particles in FND-lip and bare FND group before (**a**) and after (**b**) cell division. The correlation between particle number and cell volume is shown in panel (**c**) for FND-lip and in panel (**d**) for bare FNDs. The red line represents the cutting point between mother and daughter cells. One-hundred cells have been used for these experiments, and error bars represent standard deviation with significance level 0.05.

**Figure 5 nanomaterials-10-00516-f005:**
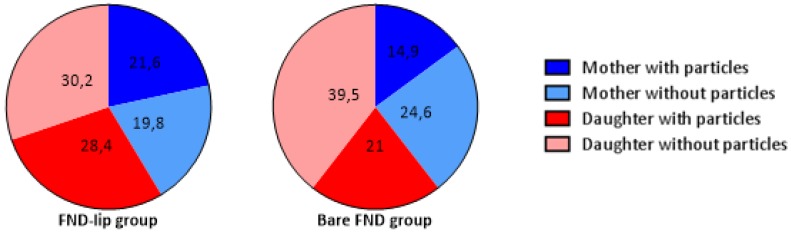
Particle distribution after cell division in yeast cells (in percentages). From the FND-lip and bare FND groups. There are more daughter cells that are containing particles than mother cells.

**Figure 6 nanomaterials-10-00516-f006:**
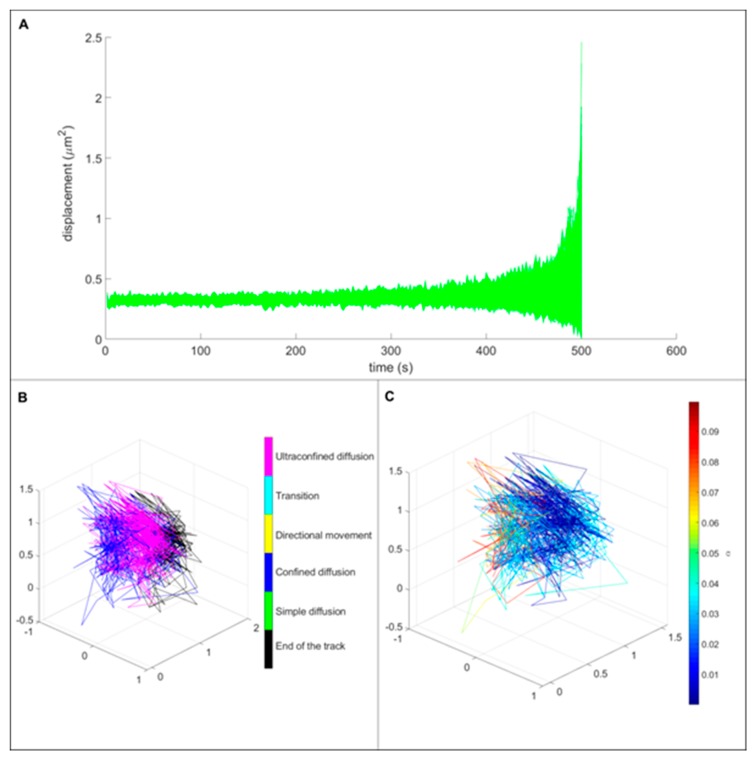
Movement of bare diamonds in cells. The mean square displacement (MSD) curves of bare FNDs on top (**A**) show only confined diffusion. Panel (**B**) shows the complete three-dimensional trajectory, showing the types of movement detected at each segment of the trajectory. The only types of movement observed were confined diffusion (blue) and ultraconfined diffusion (magenta), for which the calculated α approaches 0, and no displacement can be detected. The final part of the trajectory, for which no analysis can be performed, is shown in black. The right lower panel (**C**) shows the values of α observed at different segments of the trajectory.

**Figure 7 nanomaterials-10-00516-f007:**
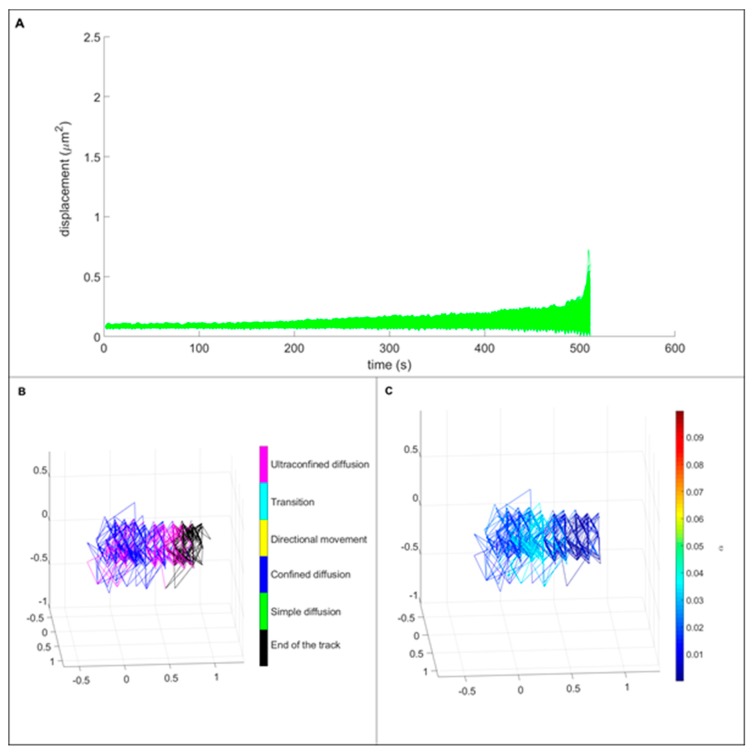
Movement of FND-lip in cells. (**A**) MSD curves of lipid-coated FNDs. MSD analysis reveals that they only move in confined diffusion and ultraconfined diffusion modes. The image on the left side of the bottom panel (**B**) is the three-dimensional trajectory, colored to show the modes of motion that have been detected at different segments of the track. Just like in bare FNDs, no other types than confined and ultraconfined diffusion were observed. The 3D trajectory on the right (**C**) shows the fluctuations of estimated α, reflecting the degree of confinement. Note that the overall volume explored by the particle is smaller, compared to the bare FNDs, resulting in lower α values.

**Figure 8 nanomaterials-10-00516-f008:**
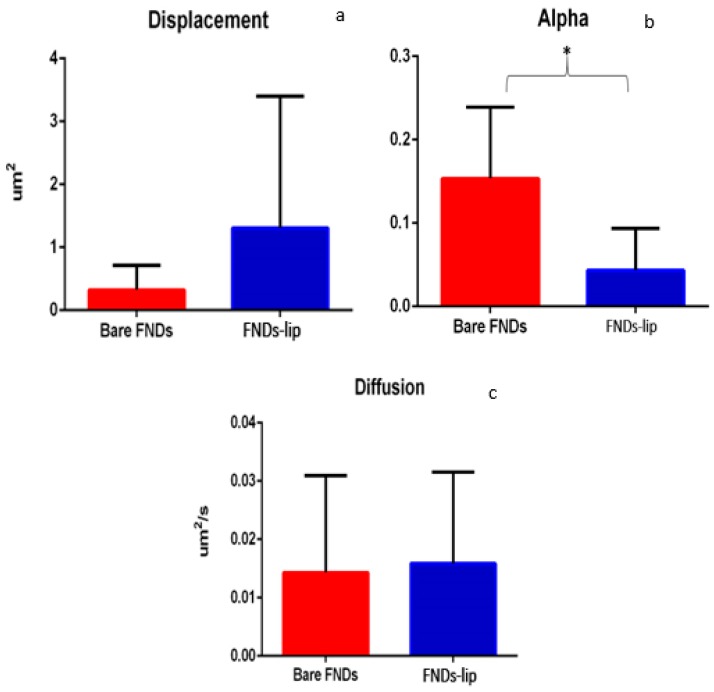
Statistical comparison between bare-FNDs and FNDs-liposome. Panel (**a**) compares the displacement of each particle. Panel (**b**) is a comparison between alphas of each particle. A significant decrease was found in the case of FND-lip. Panel (**c**) is a comparison between the diffusion coefficient. All of these experiments have been done in triplicates, error bars represent standard deviation, and statistical differences are tested at significance level 0.05. * indicates *p* ≤ 0.05.

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
