# Peer review of "The Fate of Lipid-Coated and Uncoated Fluorescent Nanodiamonds during Cell Division in Yeast"

_nanomaterials, 2020, doi:10.3390/nano10030516_

Round 1

Author Response

Dear Reviewers and Editors,

First of all we would like to thank you for your time that you have invested in improving our manuscript and off course for your positive judgement. We have addressed the comments from the reviewers and hope that you will find our article ready for publications now. Below you can find the responses to the reviewer comment.

Reviewer comments

Reviewer 1

Abstract

  • Final sentence of the abstract says, “Finally, we also compared bare nanodiamonds with lipid-coated diamonds and there were no significant in respect to both of movement or intracellular fate.” The sentence is missing a verb after the word significant.

We have corrected the error.

Materials and Methods 2.1.1:

  • How did you measure the zeta potential? Please provide machine, how long the runs were, what buffer was used, cuvettes, etc.

We added the details for these measurements to the respective section of the materials and methods.

 2.1.2:

  • It is difficult to know if the coating process simply results in empty lysosomes plus FND (the heterogenous combination of which would still alter your zeta potential), or if the FND is actually coated. Reference 18’s Figure 3 does seem to suggest that coating does happen, so it would be useful to mention this in your paper.

We added a sentence on the previous results here to make this more clear.

Results Figure 4:

  • This is a bit confusing. What is the difference between particles/cell and objects/cell?

We define an object as connected cluster particles. This was done acknowledging the fact that an object can be either a single particle or an aggregate of particles. The difference between the number of objects and the number of particles reveals the aggregation status of the sample in the intracellular environment (rather than in a test tube as in DLS for instance). We added a sentence to make this more clear.

Also, is there a way to plot the data from (a) and (b) on the same combined graph, perhaps using a log scale to help make the differences more visually apparent?

We followed the suggestion of the reviewer and combined (a) and (b) in a graph that is scaled logarithmically.

Section 3.2:

  • Without actually measuring the signal from the culture medium and completing the mass balance, one can’t say “This means that for both particle types a large proportion of diamond particles is excreted from the cells.” Without these measurements, it would be better to say you surmise that they are being secreted.

We have followed the suggestion of the reviewer and formulate this more carefully now.

  • For the sentence, “We do not find a significant difference in number of objects per cell and number of particles per cell between FND-lip and bare FNDs before cell division and after cell division (P value<0.05)”, do you mean P value > 0.05, i.e. not statistically significant?

There is no significant difference between the 2 particle types but there is between before and after. To make this more clear we have reformulated the statement.

General comment:

  • Can you add a statement (conclusion) at the end commenting on the overall significance of this work?

As suggested, we have added a comment on the potential impact of the work.

Reviewer 2 Report

  1. The authors need to prove that FND and FND-Lip particles did not bring any negative impact to the yeast cells subjected to cell wall removal. All the studies in this research based on the hypothesis that those two particles are quite biocompatible to the yeast cells and the spheroplasting protocol did not compromise their viability and proliferation, neither.
  2. If these two particles are toxic after their endocytosis, how did the author exclude those dead yeast cells in their study? If the authors did not do that, then the dead cells would be counted as mother cells, which was not true, because they would not have any daughter cells at all. And even those mother yeast cells were not dead after endocytosis of those particles, they might stop dividing, and they could be considered as mother cells, neither. They authors need to make a detailed clarification of all these.
  3. The bare FND is highly positively charged and it was very likely to be toxic to the yeast cells. The authors need to prove their biocompatibility specifically.
  4. The number of daughter yeast cells with particles should be “210” rather than “21”. Please revise the pie of the bare FND group in Figure 5 accordingly.
  5. All the pictures in the paper are not in good resolution and the authors need to re-edit them accordingly.

Author Response

Dear Reviewers and Editors,

First of all we would like to thank you for your time that you have invested in improving our manuscript and off course for your positive judgement. We have addressed the comments from the reviewers and hope that you will find our article ready for publications now. Below you can find the responses to the reviewer comment.

Reviewer comments

Reviewer 2:

  • The authors need to prove that FND and FND-Lip particles did not bring any negative impact to the yeast cells subjected to cell wall removal. All the studies in this research based on the hypothesis that those two particles are quite biocompatible to the yeast cells and the spheroplasting protocol did not compromise their viability and proliferation, neither.

This is indeed a very important point. Biocompatibility is generally very well reported for nanodiamonds. (we have added references there) Also the specific particles have already been shown to be very biocompatible. This was already demonstrated in ref 18 and thus not repeated here. But we have added a comment on the topic.

  • If these two particles are toxic after their endocytosis, how did the author exclude those dead yeast cells in their study? If the authors did not do that, then the dead cells would be counted as mother cells, which was not true, because they would not have any daughter cells at all. And even those mother yeast cells were not dead after endocytosis of those particles, they might stop dividing, and they could be considered as mother cells, neither. They authors need to make a detailed clarification of all these.

For most of this article we did follow the cells by microscopy through the division on a single cell basis and all the cells that we observed did divide.  Only in Figure 4 we did not do this to have a larger statistic. However, both the previous work (ref 18) on biocompatibility and the work on single cells suggest that this is valid.

  • The bare FND is highly positively charged and it was very likely to be toxic to the yeast cells. The authors need to prove their biocompatibility specifically.

This is a misunderstanding. Bare FNDs have a negative zeta potential.

  • The number of daughter yeast cells with particles should be “210” rather than “21”. Please revise the pie of the bare FND group in Figure 5 accordingly.

There is probably a misunderstanding here. It is correct as it is now. The data is in %. And it is indeed 21%.

  • All the pictures in the paper are not in good resolution and the authors need to re-edit them accordingly.

This probably has to do with the conversion to PDF but we have also submitted the raw figures for production.